# ForceNet: A Graph Neural Network for Large-Scale Quantum Chemistry Simulation

## Abstract

Machine Learning (ML) has a potential to dramatically accelerate large-scale physics-based simulations. However, practical models for real large-scale and complex problems remain out of reach. Here we present ForceNet, a model for accurate and fast quantum chemistry simulations to accelerate catalyst discovery for renewable energy applications. ForceNet is a graph neural network that uses surrounding 3D molecular structure to estimate per-atom forces—a central capability for performing atomic simulations. The key challenge is to accurately capture highly complex and non-linear quantum interactions of atoms in 3D space, on which forces are dependent. To this end, ForceNet adopts (1) expressive message passing architecture, (2) appropriate choice of basis and non-linear activation functions, and (3) model scaling in terms of network depth and width. We show ForceNet reduces the estimation error of atomic forces by 30% compared to existing ML models, and generalizes well to out-of-distribution structures. Finally, we apply ForceNet to the large-scale catalyst dataset, OC20. We use ForceNet to perform quantum chemistry simulations, where ForceNet is able to achieve $4\times$ higher success rate than existing ML models. Overall, we demonstrate the potential for ML-based simulations to achieve practical usefulness while being orders of magnitude faster than physics-based simulations.

## 1 Introduction

Learning models for simulating complex physical systems has attracted much recent attention (Sanchez-Gonzalez et al., 2020; Bapst et al., 2020; Kipf et al., 2018; Battaglia et al., 2016; Gilmer et al., 2017; Schütt et al., 2017; Klicpera et al., 2020). The premise is that once an accurate ML-based simulator is obtained, it can perform inference orders-of-magnitude faster than the original underlying physics-based simulator. Many existing works have focused on relatively small-scale simple domains, where physics-based simulation is cheap (*i.e.*, order of seconds), such as simulating springs and oscillators (Battaglia et al., 2016; Kipf et al., 2018), fluids and rigid solids (Sanchez-Gonzalez et al., 2020) that follow classical Newtonian dynamics, and glassy systems (Bapst et al., 2020) that follow the basic Lennard–Jones potential. Other approaches have explored complex domains with smaller systems, such as small organic molecules (Ramakrishnan et al., 2014; Schütt et al., 2017; Klicpera et al., 2020).

An important open question is whether ML-based simulation is effective in larger and more complex quantum chemistry domains. If successful, ML could be applied to problems such as catalyst discovery which is key to solving many societal and energy challenges including solar fuels synthesis, long-term energy storage, and renewable fertilizer production (Seh et al. (2017); Jouny et al. (2018); Whipple & Kenis (2010)). Currently, Density Functional Theory (DFT) (Parr, 1980) is a popular and reliable, but computationally expensive approach to performing quantum chemistry simulation tasks, such as atomic structure relaxation (Figure 1 (left)) and molecular dynamics. Approximating DFT with ML-models is an exceptionally challenging task given a simulation's sensitivity to the atoms' elements and subtle changes in the atoms' positions. However, if successful, accurate and fast ML-based models may lead to significant practical impact by accelerating simulations from $O$(hours-days) to $O$(ms-s), which in turn accelerates applications such as catalyst discovery.

A key component of any ML-model used for atomic simulations is predicting the non-linear and complex forces applied on atoms by their neighbors. These forces are highly sensitive to the element type and small changes in placement of neighboring atoms. One approach that is well-suited to

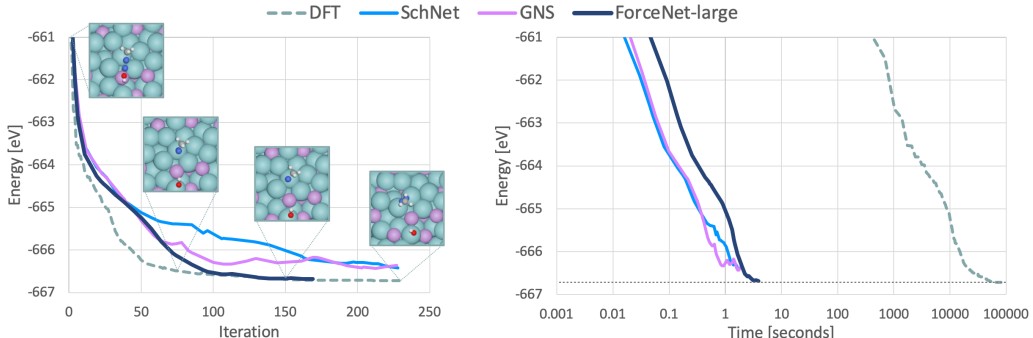

Figure 1: **(left)** Illustrative example of atomic structure energies during relaxations performed using DFT and ML-based (SchNet, GNS, ForceNet) approaches. 3D renderings of the structures along the relaxation trajectory are shown, where small atoms are adsorbates and larger atoms are catalysts. The ForceNet (ours) model reaches a low energy similar to DFT. **(right)** Same plot, but changing the x-axis to time rather than iteration steps. All ML models are more than $10^3\times$ faster (GPU acceleration of DFT is only 2-5$\times$ (Maintz & Wetzstein, 2018)), while only ForceNet finds similar energy to that of DFT in this example. Note different optimizers were used for DFT (conjugate gradient) vs. ForceNet (L-BFGS), which may have led to them reaching the same energy by different trajectories.

the modeling of local interactions is Graph Neural Networks (GNNs) (Gilmer et al., 2017) where nodes represent atoms and messages passed along edges represent the atom interactions. In this paper, we present ForceNet that demonstrates a significant improvement in quantum simulation performance, and provides strong evidence that GNN-based simulation is effective in practically-relevant and realistic domains. We build on the recent framework of Graph Network-based Simulators (GNS) (Sanchez-Gonzalez et al., 2020; Bapst et al., 2020), where node movements (*i.e.*, atomic forces in our application) are predicted from the node embeddings of a GNN. We address the key challenge of accurately capturing the local 3D atomic structure using three approaches:

**Conditional filter convolution:** To model the complex atom interactions in 3D space, we extend the continuous filter convolution (Schütt et al., 2017) to make it much more expressive; we condition the convolution filter not only on the Euclidean distance between atoms, but also on the embeddings of source and target nodes as well as the $(x, y, z)$ directional differences.

**Carefully-chosen basis and activation functions:** We demonstrate that it is critical to carefully select the appropriate non-linear activation and basis function in the message-passing architecture, so that complex non-linear dynamics can be effectively captured. Through comprehensive empirical studies, we identify a particularly effective design choice that is based on spherical harmonics and Swish activation (Ramachandran et al., 2017).

**Scaling:** The scaling of model size is necessary for capturing the complexity of the forces. The model is scaled with respect to depth and width to significantly improve its performance.

We apply our model to the new large-scale quantum chemistry dataset OC20 (Anonymous, 2020) that contains 200+ million samples from atomic relaxations relevant to the discovery of new catalysts for renewable energy storage and other energy applications. The result of our work is a simple and scalable ForceNet model that achieves state-of-the-art performance in predicting quantum force fields, reducing the MAE force errors by 30% compared to existing GNN models. Similar performance gains are obtained on out-of-distribution tasks for which similar atomic structures are not seen during training. Finally, we use ForceNet as a surrogate to DFT to perform quantum chemistry simulation; specifically, calculating structure relaxations of complex systems, a task of high practical value. We show that ForceNet is able to estimate the relaxed structures $4\times$ more accurately than previous state-of-the-art, while being multiple-orders of magnitude faster than DFT, Figure 1.

## 2    RELATED WORK

**Message-passing GNNs.**    Our ForceNet is based on message passing GNNs (Gilmer et al., 2017) that iteratively update node embeddings based on messages passed from neighboring nodes. In its most general form, the message function depends on the two node embeddings as well as edge

features. Many GNN variants fall under this framework (Kipf & Welling, 2017; Velickovic et al., 2018; Xu et al., 2019). The recently-proposed GNN-FiLM (Brockschmidt, 2020) uses an embedding of the target node to modulate the message from the source nodes, and is closely related to our use of a conditional filter. However, most existing message-passing GNNs, including GNN-FiLM, are designed for homogeneous graphs without edge features. Consequently, edge features are often incorporated in an adhoc manner and can be ineffective at approximating complex atomic interactions, which are central to this work.

**Force-centric Graph Network-based Simulation (GNS).** Our ForceNet builds on the force-centric GNS framework (Sanchez-Gonzalez et al., 2020; Bapst et al., 2020). Here a model's primary output is per-atom forces (thus, force-centric), which are then used to perform simulation, *i.e.*, updating atom positions based on forces. The GNS framework follows the three steps to predict forces: (1) A graph is constructed from 3D points, (2) an encoder GNN is applied to the graph to obtain node embeddings, and (3) a decoder is applied to the node embedding to predict the per-node forces. The GNS framework has demonstrated promising results on relatively simple domains such as fluid dynamics and glassy systems, where ground-truth simulation is already cheap and can be generated on-the-fly during training. Compared to these domains, using ML to replace expensive quantum mechanical simulation is more practically impactful and challenging. In fact, as we demonstrate empirically, the off-the-shelf GNS model fails to accurately predict the quantum mechanical forces.

**Energy-centric Simulation.** The majority of GNN models developed for quantum chemistry simulation fall under the energy-centric simulation framework, in which a model's primary output is the energy of the entire atomic system (hence, energy-centric). The atomic forces are then derived implicitly through negative gradients of energy with respect to the atomic positions, which guarantees the force-field is energy-conserving. Although physically-principled, energy-centric models are empirically found to underperform the force-centric models (especially ForceNet and its variants) in our task. This is possibly because the force-centric models capture the dependency of atomic interactions on atomic forces more explicitly than the energy-centric models.

Many sophisticated GNN architectures have been proposed under the energy-centric framework, such as SchNet (Schütt et al., 2017) and DimeNet (Klicpera et al., 2020). Our conditional filter convolution is built from SchNet's *continuous filter convolution* architecture, where we make an important extension to resolve a number of critical issues when adopting it to the force-centric GNS framework (see Section 3.1.1 for details).

## 3 ForceNet

Here we introduce ForceNet by specifying its input and output, the model architecture and training strategy. The goal of the OC20 dataset is to take a set of 3D atom positions, referred to as the initial structure, and find its relaxed structure. The relaxed structure is defined as a set of 3D atom positions that correspond to a local energy minimum for the system. This is typically accomplished by computing the per-atom forces, updating the atom positions based on the forces, and repeating until the forces approach zero. Our goal is to compute these forces using ForceNet. Additional details about the OC20 dataset can be found in the supplementary material.

The input to ForceNet is an atomic structure, *i.e.*, a set of $M$ atoms and their 3D spatial positions. The output is a 3D vector for each node, representing the predicted $(x, y, z)$ atomic force.

### 3.1 Model architecture

ForceNet represents atoms as nodes in a GNN and the atomic interactions as edges. The node input features specify the atom's atomic number and other properties (9-dimensional vector adapted from Xie & Grossman (2018)). Edges in the GNN are constructed from a radius graph of neighboring atoms (Schütt et al., 2017; Sanchez-Gonzalez et al., 2020). Let $\mathcal{N}_t(c)$ denote a set of neighboring atoms that are within the cutoff-distance $c$ away from the target atom $t$. On average an atom has 35 neighbors. A directed edge $e_{st}$ from source atom $s$ to target atom $t$ is drawn for $s \in \mathcal{N}_t(c)$. Let $\boldsymbol{d}_{st} \in \mathbb{R}^3$ be their relative displacement, *i.e.*, a vector pointing from atom $s$ to atom $t$.

ForceNet follows the encoder-decoder architecture of the GNS framework (Sanchez-Gonzalez et al., 2020; Battaglia et al., 2016; Kipf et al., 2018). The encoder uses iterative message passing to compute node embeddings $\boldsymbol{h}_t$ that capture the 3D structure surrounding each atom, and the decoder uses an

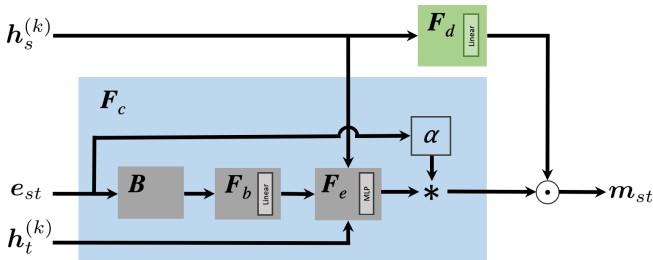

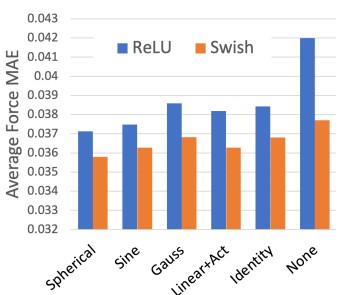

Figure 2: Model diagram for messages $m_{st}$ used by ForceNet in Eqns. (2) and (3). Two key components are (1) the expressive conditional filter $F_c$ that is dependent on full edge feature as well as source and target node embeddings, and (2) the basis function $B$ over the edge feature that helps the network to accurately capture atomic interactions.

Figure 3: Ablations on basis functions (x-axis) and activation functions (ReLU and Swish) in ForceNet.

MLP to directly predict per-atom forces from these embeddings. The encoder updates $h_t$ as:

$$h_t^{(k+1)} = F_n\left(m_t + \sum_{s \in \mathcal{N}_t} m_{st}\right) + h_t^{(k)}, \tag{1}$$

where the messages $m_{st}$ and $m_t$ are summed and passed through the function $F_n : \mathbb{R}^D \to \mathbb{R}^D$ that is a 1-hidden-layer MLP with batch normalization (Ioffe & Szegedy, 2015). The dimensionality of the node and hidden layer features is $D$. Equation (1) follows standard GNN embedding update formulations (Gilmer et al., 2017) with the addition of a residual connection, $h_t^{(k)}$ (He et al., 2016). We define the directional messages $m_{st}$ and self message $m_t$ in the next section.

The decoder is computed using the last layer $K$'s node embeddings $h_t^{(K)}$, $f_t = F_f(h_t^{(K)})$ where $f_t$ is the 3D force of atom $t$, and $F_f$ is a 1-hidden-layer MLP with batch normalization.

The critical aspect of ForceNet is its encoder and specifically the message computation that effectively captures the non-linear and complex atomic interactions to predict the atomic forces. To this end, we present our approach to message computation and its three key architectural components.

### 3.1.1 Conditional Filter Convolution

The computation of messages in ForceNet utilizes a simple yet effective extension of the continuous filter convolution (Schütt et al., 2017). The continuous filter convolution of (Schütt et al., 2017) passes the distance between neighboring atoms through a set of radial basis functions to create an embedding used to compute the messages. This approach has the following limitations that arise when transitioning from an energy-centric to a force-centric model: (1) edge features are only computed from atom distance information, i.e., 3D angular information is lost. This results in the node embeddings being rotation-invariant, but forces are rotation-covariant (rotate together with a molecular system) (2) the filter does not take into account information about the node or atom properties. Changes in atom properties, such as atomic number and radii, may result in significant force differences for atoms at similar distances.

We resolve these issues by using our $E$-dimensional edge feature $e_{st}$ (described below) that encodes rotation-covariant directional information, and conditioning on both the source $h_s^{(k)}$ and target $h_t^{(k)}$ node information:

$$m_{st} = F_c(h_s^{(k)}, e_{st}, h_t^{(k)}) \odot F_d(h_s^{(k)}), \tag{2}$$

where $F_c : \mathbb{R}^D \times \mathbb{R}^E \times \mathbb{R}^D \to \mathbb{R}^D$ is the conditional filter, and $F_d : \mathbb{R}^D \to \mathbb{R}^D$ is a learnable linear function. The edge feature $e_{st} \equiv \text{Concat}(n_{st}, p_{st}/c) \in \mathbb{R}^7$, where $n_{st} \equiv d_{st}/\|d_{st}\|$ is a normalized directional vector and $p_{st} \in \mathbb{R}^4$ is a list of four atomic distances $\|d_{st}\|, \|d_{st}\| - a_s, \|d_{st}\| - a_t, \|d_{st}\| - a_s - a_t$ that take into account the atomic radii $a_s$ and $a_t$ of atoms $s$ and $t$, respectively. While the use of $e_{st}$ results in the model not being rotation invariant due to the use of $n_{st}$, data augmentation may be used to encourage rotation-covariance of the model's predictions (explored further in Section 3.2). As our results demonstrate, applying the conditional filter $F_c$ on the

embeddings $\boldsymbol{F}_d$ of neighboring nodes through the element-wise dot product significantly improves performance, even though $\boldsymbol{F}_c$ already contains the source node information $\boldsymbol{h}_s^{(k)}$.

The conditional filter $\boldsymbol{F}_c$ combines the raw edge features $\boldsymbol{e}_{st}$ with the node embeddings to encode the interactions between atoms $s$ and $t$ (Figure 2) and is defined as :

$$\boldsymbol{F}_c(\boldsymbol{h}_s^{(k)}, \boldsymbol{e}_{st}, \boldsymbol{h}_t^{(k)}) = \alpha(\|\boldsymbol{d}_{st}\|) \cdot \boldsymbol{F}_e\left(\boldsymbol{h}_s^{(k)}, \boldsymbol{F}_b(\boldsymbol{B}(\boldsymbol{e}_{st})), \boldsymbol{h}_t^{(k)}\right), \tag{3}$$

where $\alpha(x) = \cos(\pi x / 2c)$ is a scalar that decays to zero as $\|\boldsymbol{d}_{st}\|$ approaches the distance cutoff $c$. $\boldsymbol{F}_e$ is a 2-hidden-layer MLP with the hidden size of $D$ and the three input vectors are concatenated as input. $\boldsymbol{B} : \mathbb{R}^E \to \mathbb{R}^B$ is the basis function we discuss in the next section, and $\boldsymbol{F}_b : \mathbb{R}^B \to \mathbb{R}^D$ is a learnable linear function that maps the $B$ dimensional vector to a $D$ dimensional vector for input to $\boldsymbol{F}_e$. The parameters for $\boldsymbol{F}_b$ are shared across layers, while no other parameters are shared across layers. Finally, the self message $\boldsymbol{m}_t$ is defined by applying an element-wise product between learnable filter $\boldsymbol{v} \in \mathbb{R}^D$ and $\boldsymbol{F}_d$, *i.e.*, $\boldsymbol{m}_t = \boldsymbol{v} \odot \boldsymbol{F}_d(\boldsymbol{h}_t^{(k)})$.

### 3.1.2 Basis Functions

An important aspect of $\boldsymbol{F}_c$ is the choice of basis function $\boldsymbol{B} : \mathbb{R}^E \to \mathbb{R}^B$ that transforms the raw distance features $\boldsymbol{e}_{st}$ into ones that are more discriminative. Several choices of basis functions have been proposed, such as a Gaussian over 1D distances (Schütt et al., 2017) and a spherical Bessel function over the joint 2D space of the edge distance and angle (Klicpera et al., 2020). We extend these ideas to capture the full 3D positional differences between atoms, and systematically study the effectiveness of different basis functions in the context of a force-centric model.

Each of our basis functions $\boldsymbol{B}$ maps the raw edge distance features $\boldsymbol{e}_{st}$ into a $B$ dimensional vector, where $B$ varies based on the basis function used. $B$ is typically much larger than $E = 7$ to aid in capturing subtle differences in atom positions.

**Identity:** $\boldsymbol{B}_{\text{id}}(\boldsymbol{x}) = \boldsymbol{x}$. The baseline is to use the edge features $\boldsymbol{e}_{st}$ directly.

**Linear + Act:** $\boldsymbol{B}_{\text{linact}}(\boldsymbol{x}) = g(\boldsymbol{W}\boldsymbol{x} + \boldsymbol{b})$, where $g(\cdot)$ is the non-linear activation function, and $\boldsymbol{W}$ and $\boldsymbol{b}$ are the learnable parameters. When followed by the linear layer $\boldsymbol{F}_b$, this is equivalent to applying an 1-hidden-layer MLP over the edge features $\boldsymbol{e}_{st}$.

**Gaussian:** $\boldsymbol{B}_{\text{gauss}}(\boldsymbol{x}) = [b_1, \ldots, b_J]$, where $b_j$ is the output of the $j$-th basis function $b_j(x) = \exp^{(x - \mu_j)^2 / (2 \cdot \sigma^2)}$. The Gaussian means are evenly distributed on the interval between 0 and 1, *i.e.*, $\mu_j = j / (J - 1)$ and the standard deviation $\sigma = 1 / (J - 1)$. All values of $x$ are normalized to lie between 0 and 1. $\boldsymbol{B}_{\text{gauss}}$ is applied to each dimension of $\boldsymbol{e}_{st}$, resulting in a $B = J \times E$ vector.

**Sine:** $\boldsymbol{B}_{\sin}(\boldsymbol{x}) = [b_1, \ldots, b_J]$, where $b_j$ is the output of the $j$-th basis function $b_j(x) = \sin(1.1^j x)$. The design is based on function approximation using the Fourier series. In our experiments, we find that using only the sinusoidal component of the Fourier series is sufficient. $\boldsymbol{B}_{\sin}$ is applied to each dimension of $\boldsymbol{e}_{st}$, resulting in an $B = J \times E$ vector.

**Spherical harmonics:** $\boldsymbol{B}_{\text{sph}}(\boldsymbol{e}_{st}) = \boldsymbol{Y}_L(\theta, \phi)\boldsymbol{R}(\boldsymbol{p}_{st})^\top$ where $\boldsymbol{Y}_L$ is the list of Laplace's spherical harmonics (MacRobert, 1947) used to encode the angular information and $\boldsymbol{R}(\boldsymbol{p}_{st})$ encodes the distance. We use spherical harmonic functions up to degree $L$, which gives us $L^2$ orthogonal basis in total. The angles $\theta$ and $\phi$ can be directly computed from $\boldsymbol{n}_{st} \subset \boldsymbol{e}_{st}$. $\boldsymbol{R}$ uses a linear combination of the above sine basis functions computed from $\boldsymbol{p}_{st} \subset \boldsymbol{e}_{st}$ (thus, $4J$ basis functions in total) to encode distance information. Specifically, $\boldsymbol{R}(\boldsymbol{p}_{st}) = \boldsymbol{W}_{\text{rad}}\boldsymbol{B}_{\sin}(\boldsymbol{p}_{st}) + \boldsymbol{b}_{\text{rad}} \in \mathbb{R}^S$, where $\boldsymbol{W}_{\text{rad}}$ and $\boldsymbol{b}_{\text{rad}}$ are learnable parameters. $\boldsymbol{B}_{\text{sph}}$ is flattened into a vector before being passed into $\boldsymbol{F}_b$. The dimensionality of $\boldsymbol{B}_{\text{sph}}$ is $B = SL^2$, where we set $S$ to be the dimensionality of $\boldsymbol{p}_{st}$.

### 3.1.3 Expressive Non-Linearity in MLPs

Our final key component is simple: the choice of non-linear activation function plays a central role in modeling complex non-linearities. The popular ReLU activation is not ideal, since it results in outputs being modeled as piece-wise linear hyper-planes with sharp boundaries. In contrast, atomic forces are locally smooth with continuous curvatures. Furthermore, negative values are set to zero in ReLU, reducing its expressivity. Ideally, we desire a smooth and expressive non-linear activation function. Here, we adopt the Swish activation, *i.e.*, $\text{act}(x) = x \cdot \text{sigmoid}(x)$ (Ramachandran et al., 2017), which provides a smoother output landscape and has non-zero activation for negative inputs.

As we demonstrate in Section 4, the replacement of ReLU with Swish consistently and significantly improves the predictive accuracy while maintaining scalability across all choices of basis functions.

## 3.2 Training Strategies

We present two strategies for training ForceNet. First, we apply random rotation data augmentation to encourage the rotation-covariance of the model. In our application, the $z$-axis is canonicalized and is always vertical to the material surface. Thus, we only need to rotate the systems and the forces along the $z$-axis. Second, during the relaxation process, some atoms are kept fixed while others are free to move. At test time, evaluation is only performed on free atoms. However, we train on both free and fixed atoms using different weights for the loss. See Table 7 in the supplementary material.

## 4 Experiments

In this section, we show that ForceNet significantly improves on predicting forces compared to both energy-centric GNNs (Schütt et al., 2017; Klicpera et al., 2020) and existing force-centric GNS models (Sanchez-Gonzalez et al., 2020). Through a series of ablation studies, we provide insight into how each building block of ForceNet and model size contributes to its performance. We also use ForceNet to perform structure relaxations and demonstrate its practical effectiveness.

**Experimental settings.** We train and evaluate our models on a new large-scale quantum chemistry simulation dataset, OC20 (Anonymous, 2020), that is relevant to the discovery of new catalysts for renewable energy storage and other energy applications. The dataset includes 200+ million simulation samples from 1M+ atomic relaxations containing large atomic structures (20-200+ atoms) and corresponding per-atom forces. Compared to the existing quantum chemistry QM9 dataset (Ramakrishnan et al., 2014), OC20 contains three-orders-of-magnitude more data points and was constructed from 70 million hours of highly accurate Density Functional Theory (DFT) calculations.

Each structure contains atoms corresponding to a catalyst's surface and an adsorbate. A catalyst is a material used to increase the reaction rate and efficiency of a chemical reaction, and an adsorbate is a molecule involved in the chemical reaction that interacts with the catalyst. We evaluate models on four validation datasets that test different levels of model generalization: In Domain (ID), Out of Domain Adsorbate (OOD Adsorbate), OOD Catalyst, and OOD Both (both the adsorbate and catalyst's material are not seen in training). Each split contains 1M examples. The Mean Absolute Error (MAE) of per-atom forces on free atoms is evaluated for each validation set. We will evaluate our models on the hidden test sets once that data is made available.

For basis and activation functions, we use the spherical function and Swish activation by default, as this combination consistently provides the best results (see Figure 3, and Table 5 in Appendix D). For all experiments on force-centric GNNs (including the GNS model), we adopt the two strategies presented in Section 3.2, *i.e.*, the rotation augmentation and loss reweighting. In Appendix D, we perform ablation studies and show that both strategies are helpful. Our implementation is based on Pytorch (Paszke et al., 2019) and Pytorch Geometric (Fey & Lenssen, 2019), and will be open-sourced. All the implementation details and hyper-parameters are provided in Appendix C.

**Performance comparison with existing models.** In Table 1, we compare ForceNet with existing GNN models: the energy-centric GNN models, SchNet (Schütt et al., 2017) and DimeNet (Klicpera et al., 2020), are as reported in (Anonymous, 2020). Note that DimeNet performs message passing over triplets of nodes; thus, its model size was reduced to make training tractable. The GNS models are adopted from Sanchez-Gonzalez et al. (2020) based on feedback from the original author (see Appendix B for implementation details), where we set its model size to be comparable to ForceNet. The trivial baseline of always predicting force medians for all atoms is added to the table for reference.

The difficulty of the task is apparent in the errors of existing models. In all validation splits, the models' force MAEs are often far more than even half of that produced by the trivial median baseline. Contrary to the results in Klicpera et al. (2020), we observed that DimeNet performs worse than SchNet on OC20. We believe this is due to limits in considering all atom triplets for angular terms in the node updates, which are harder to fit in memory with growing graph and model sizes.

ForceNet models reduce the error significantly in both ID and OOD scenarios. Compared to the best existing model (the GNS model), our best ForceNet-large reduces the average MAE by 0.0139 eV/Å (29.5% relative improvement). Comparing models with similar numbers of parameters we see that ForceNet (with 11.4M parameters) reduces the average MAE of the GNS model (with 12.5M

Table 1: Comparison of ForceNet to existing GNN models. We mark as bold the best performance and close ones, *i.e.*, within 0.0005 MAE, which according to our preliminary experiments, is a good threshold to meaningfullly distinguish model performance. The final row represents the relative error reduction of our best model (ForceNet-large) compared to the best existing model (GNS).

| Model | Width | Depth | #Params | Validation Force MAE (eV/Å) | | | | |
|---|---|---|---|---|---|---|---|---|
| | | | | ID | OOD Ads. | OOD Cat. | OOD Both | Average |
| Median | – | – | – | 0.0757 | 0.0743 | 0.0744 | 0.0868 | 0.0778 |
| DimeNet (Klicpera et al., 2020) | 256 | 3 | 4.4M | 0.0600 | 0.0579 | 0.0575 | 0.0702 | 0.0614 |
| SchNet (Schütt et al., 2017) | 1024 | 4 | 7.4M | 0.0438 | 0.0511 | 0.0458 | 0.0619 | 0.0507 |
| GNS (Sanchez-Gonzalez et al., 2020) | 768 | 5 | 12.5M | 0.0419 | 0.0469 | 0.0430 | 0.0562 | 0.0470 |
| **ForceNet** | 512 | 5 | 11.3M | 0.0314 | 0.0348 | 0.0336 | 0.0433 | 0.0358 |
| **ForceNet-large** | 768 | 7 | 34.8M | **0.0281** | **0.0316** | **0.0318** | **0.0410** | **0.0331** |
| **Relative error reduction of ForceNet-large** | | | | **32.9%** | **32.7%** | **26.0%** | **27.1%** | **29.5%** |

Table 2: Ablations on the architecture of ForceNet.

| Ablation | Avg Force MAE (eV/Å) |
|---|---|
| **ForceNet** | **0.0358** |
| (1) Only-dist | 0.0698 |
| (2) No-atomic-radii | 0.0368 |
| (3) No-node-emb | 0.0412 |
| (4) Only-$F_c$ | 0.0378 |
| (5) Edge-linear-BN | 0.0427 |
| (6) Node-linear-BN | **0.0357** |

Table 3: The effect of model width $D$, depth $K$ on ForceNet. All models are trained with a default batch size of 256, except the last row that uses 512.

| Width | Depth | Validation Force MAE (eV/Å) | | | | |
|---|---|---|---|---|---|---|
| | | ID | OOD Ads. | OOD Cat. | OOD Both | Average |
| 512 | 5 | 0.0314 | 0.0348 | 0.0336 | 0.0433 | 0.0358 |
| 768 | 5 | 0.0302 | 0.0338 | 0.0330 | 0.0427 | 0.0349 |
| 512 | 7 | 0.0304 | 0.0342 | 0.0330 | 0.0429 | 0.0351 |
| 768 | 7 | 0.0296 | 0.0335 | 0.0325 | 0.0422 | 0.0345 |
| **768** | **7** | **0.0281** | **0.0316** | **0.0318** | **0.0410** | **0.0331** |

parameters) by 0.0112 eV/Å (23.9% relative improvement). We also trained a shallower (depth of 3) variant of ForceNet with 7.1M parameters for comparison against SchNet (7.4M parameters). The average MAE is reduced by 0.0109 eV/Å (21.5% relative improvement) over SchNet.

**Effect of basis and activation functions.** We systematically study how choices of different basis and non-linear activation functions affect the model performance. We also compare with the "None" baseline, which does not use a basis function and directly concatenates the input edge feature into the node embeddings. In Figure 3, we see that the combination of spherical basis and Swish activation performs the best. In Tables 5 and 6 of Appendix D, we provide the full results, where we observe the consistent trends across the four validation sets. For comparison, the GNS model uses no basis function ("None") and ReLU, see Appendix B for full GNS model details.

**Effect of architecture design.** Next, we study the architectural building blocks of our conditional-filter-based message passing with fixed basis and activation functions. We consider six cases: **(1) Only-dist**: we remove $n_{st}$ from the input edge feature, i.e., $e_{st} \equiv p_{st}$, resulting in the edge features being rotation invariant. **(2) No-atomic-radii**: we set the input edge features to $e_{st} \equiv$ Concat $(n_{st}, \|d_{st}\|)$ (atomic radii information is dropped), **(3) No-node-emb**: filter $F_c$ is a function of only $e_{st}$ (conditioning on source and target node embeddings $h_s^{(k)}, h_t^{(k)}$ is dropped), **(4) Only-$F_c$**: Filter is directly aggregated, i.e., $m_{st} = F_c$, and self-message $m_t$ is omitted. **(5) Edge-linear-BN**: $F_e$ is replaced with a linear function followed by batch normalization, **(6) Node-linear-BN**: $F_n$ is replaced with a linear function followed by batch normalization. Note that in **(5)** and especially **(6)**, we find it critical to add batch normalization after the linear function to facilitate training.

Table 2 shows six ablation studies. Most notably, **(1)** is significantly worse than the rest, because rotation-invariant node embeddings are insufficient for predicting rotation-covariant forces. We also see from **(2)** and **(3)** that making the filter less expressive, especially by dropping the dependency on node embeddings, significantly hurts performance. The improvement from element-wise product parameterization $F_c \odot F_d$ is demonstrated in **(4)**. From **(5)**, we see that it is critical to utilize non-linear models for edge features, as atomic forces are highly dependent on their subtle changes, but non-linearities are not essential for node features **(6)**. Overall, our analysis suggests that ForceNet benefits most from its expressive edge computation via the conditional filter, which is directly responsible for accurately encoding the 3D neighborhood structure on which the atomic forces depend.

Table 4: Percentage of relaxations that are successful, as measured by the Average Force below Threshold (AFbT) and Average Distance within Threshold (ADwT) for the same models as in Table 1. The higher, the better. The final row represents the relative performance improvement of our best model compared to the best existing models.

| Model | AFbT (%) | | | | | ADwT (%) | | | | |
|---|---|---|---|---|---|---|---|---|---|---|
| | ID | OOD Ads. | OOD Cat. | OOD Both | Average | ID | OOD Ads. | OOD Cat. | OOD Both | Average |
| DimeNet | 0.00 | 0.00 | 0.00 | 0.00 | 0.00 | 10.94 | 9.80 | 8.36 | 10.21 | 9.82 |
| SchNet | 6.17 | 3.38 | 3.40 | 2.44 | 3.84 | 35.68 | 34.11 | 29.42 | 33.80 | 33.23 |
| GNS | 2.96 | 1.67 | 2.01 | 2.15 | 2.20 | 39.65 | 38.35 | 35.05 | 40.37 | 38.30 |
| **ForceNet** | 12.49 | 11.21 | 8.75 | 8.70 | 10.34 | 51.43 | 49.36 | 47.66 | 54.17 | 50.59 |
| **ForceNet-large** | **19.74** | **16.06** | **13.78** | **13.30** | **15.82** | **54.32** | **52.17** | **49.68** | **58.38** | **53.47** |
| **Relative Improvement** | **219.9%** | **375.1%** | **405.3%** | **445.1%** | **312.0%** | **37.0%** | **36.0%** | **41.7 %** | **44.6%** | **39.6%** |

**Effect of model size.** We try different model sizes for ForceNet, in terms of both depth $K$ and width $D$. As shown in Table 3, simply increasing the scale of the model improves performance significantly ($0.0314 \rightarrow 0.0296$ MAE on val ID). We also train a variant of our largest model ($\sim$35M parameters, 3.5 days to train on 64 GPUs) with a doubled batch size (512), and observe further gains ($0.0296 \rightarrow 0.0281$ MAE on val ID). While further scaling may lead to further performance gains, it increases inference time significantly. ForceNet-large is $\sim$3 times slower than the base model.

**Are ML models practically useful for approximating DFT structure relaxations?** We use ForceNet, as well as GNS, DimeNet, and SchNet to perform ML relaxations (replacing the DFT computed forces with those computed using the ML models). The goal is to find relaxed structure of molecules, *i.e.*, 3D structure with minimum energy. Qualitatively, Figure 1 demonstrates that ForceNet is able to successfully perform relaxations orders-of-magnitude faster than the ground-truth DFT-based simulation. Individual DFT relaxations in OC20 take numerous hours on multi-core CPUs, while ML models take 1.36 (SchNet), 1.23 (GNS), 1.70 (ForceNet) and 3.81 (ForceNet-large) seconds per relaxation to perform on a single GPU ($> 10^3 \times$ faster). DFT can be made faster with GPUs, but speed ups are significantly less (2-5x (Maintz & Wetzstein, 2018)).

Quantitatively, we use the same set of evaluation metrics as Anonymous (2020). The ML-relaxed structures are evaluated with DFT to determine whether they represent true relaxed structures. If a true relaxed structure is found, the DFT-computed forces on it should all be close to zero (local energy minimum). Specifically, we consider the strict metric—the percentage of relaxed structures where *all* the per-atom DFT computed forces are below a Threshold (FbT). The forces of the ML-predicted relaxed structures (500 from each val split, 2000 total) are evaluated with DFT (see Appendix E for the detailed procedure). We report the average FbT over thresholds ranging from 0.01 to 0.4 eV/Å. From Table 4, we see that on average, our best model (ForceNet-large) is roughly $4.1\times$ more accurate (relative improvement of 312.0%) than SchNet and $7.2\times$ more accurate (relative improvement of 618.1%) than the GNS model. For an FbT at a strict threshold of 0.05, an indication of practical usefulness, all existing models score 0%, while ForceNet shows a small and very modest success (0.4% on validation ID dataset). Following Anonymous (2020), we also report a surrogate metric that does not require expensive DFT calculations, *i.e.*, the percentage of structures with atom distances within a threshold of the ground truth averaged across thresholds (ADwT, 0.01 to 0.5Å). From Table 4, we see the relative improvements in ADwT is 39.6% on average.

## 5 CONCLUSION

Cheap, durable and efficient catalysts are an essential component to addressing the world's rising energy needs while reducing climate change. The efficient approximation of DFT for use in atomic structure relaxations has the potential to significantly increase the rate at which new catalysts can be discovered. Even moderate improvements in accuracy can lead to significant simulation efficiency gains, since ML relaxations are often 1000's of times faster than traditional DFT relaxations.

In conclusion, we demonstrate that a force-centric GNN model with carefully designed message passing can model complex atomic interactions. A significant improvement in accuracy is achieved over prior results and the results are promising when applied to structure relaxations. Given the size of the OC20 dataset, further model scaling may show further improvement, as well as, incorporating more domain specific information in the model architecture.

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

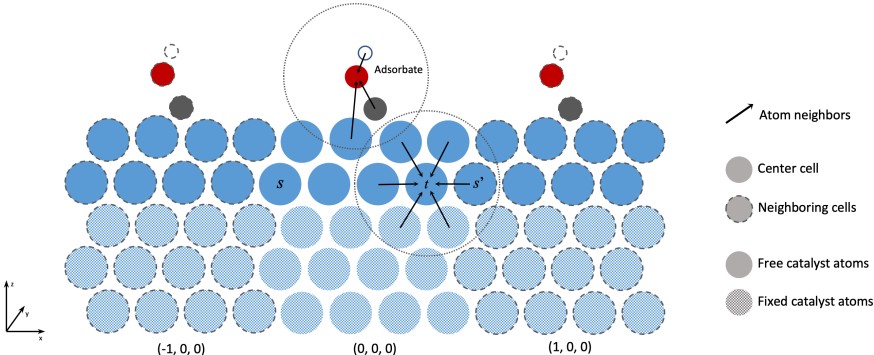

Figure 4: 2D Illustration of a slab that represents a catalyst's surface and an adsorbate. The slab is tiled in the $x$ and $y$ directions to create the surface (neighboring cells shown as atoms with dashed outlines). Only the cells to the left ($[-1, 0, 0]$) and right ($[1, 0, 0]$) are shown. The adsorbate is also assumed to be tiled with the slab (white, red, and grey atoms). Only the top 2 layers of the slab are allowed to move during a relaxation (dark blue), and the others are fixed (light blue). Neighboring atoms (black arrows) can be from the same cell or neighboring cells ($t$ and $s'$). All atoms within a radius (dotted circle) are assumed to be neighbors.

## A    DESCRIPTION OF OC20 DATASET

The OC20 dataset contains over 130 million atom structures for training. The structures are from the trajectories (the movement of the atoms from the initial state to relaxed state) contained in over 650,000 relaxations. In practical scenarios, the relaxed structure is useful in determining a chemical reaction's rate, *i.e.*, whether it will produce enough of the desired product to be useful. When applied to energy applications, such as storage, efficient methods for estimating reaction rates could have significant impact on energy scarcity and climate change.

Each structure contains the 3D positions of atoms in an adsorbate and catalyst slab, Figure 4. The adsorbate is a molecule involved in the chemical reaction that interacts with the catalyst's surface. The adsorbate contains 1 to 11 atoms. The catalyst is represented as a "slab" that repeats infinitely in the $x$ and $y$ directions. The slab structure is repeated in a grid pattern where each repetition is referred to as a "cell". The center cell has coordinate $(0, 0, 0)$ with the cell to the left right being $(-1, 0, 0)$ and $(1, 0, 0)$ respectively. The slab is not repeated in the $z$ direction. Instead, the atoms at the bottom of the slab are assumed to be fixed and not move during a relaxation, which approximates how they would be held in place by the catalyst's atoms below the slab. Typically, only the top two layers of the catalyst's surface are assumed to be free and are moved according to their forces during a relaxation (see Figure 4). Therefore, forces are only evaluated on free catalyst atoms and the adsorbate.

The forces in the same atom in different cells is assumed to be equal, since the atom neighbors are also identical. Thus, we can assume the GNN node hidden states are also identical. When computing the neighborhood of an atom, atoms in neighboring cells need to be taken into consideration (atom $t$ in Figure 4). The node embedding of atoms marked $s$ and $s'$ in Figure 4 are the same. However, the messages $\boldsymbol{e}_{st}$ and $\boldsymbol{e}_{s't}$ are different since the distance between $t$ and $s$ is not the same as $t$ and $s'$. Computing the distances between atoms from different cells can be done using the supplied information in the OC20 dataset for periodic boundary conditions.

## B    DETAILS OF GNS MODEL

For the results in this paper, we reimplemented the original GNS model (Sanchez-Gonzalez et al., 2020). Since the public code for the original GNS model (Sanchez-Gonzalez et al., 2020) was not available at the time of our experiments, we communicated with one of the authors to confirm the implementation details.

The message in a GNS model is defined as

$$m(\boldsymbol{h}_t^{(l)}, \boldsymbol{e}_{st}, \boldsymbol{h}_s^{(l)}) = \textbf{MLP}\left(\text{Concat}\left(\boldsymbol{h}_t^{(l)}, \boldsymbol{e}_{st}, \boldsymbol{h}_s^{(l)}\right)\right),$$

Table 5: Ablations on basis and activation functions in the ForceNet architecture.

| Basis | Activation | Validation Force MAE (eV/Å) | | | | |
| --- | --- | --- | --- | --- | --- | --- |
| | | ID | OOD Ads. | OOD Cat. | OOD Both | Average |
| Spherical | ReLU | 0.0323 | 0.0366 | 0.0344 | 0.0452 | 0.0371 |
| **Spherical** | **Swish** | **0.0314** | **0.0348** | **0.0336** | **0.0433** | **0.0358** |
| Sine | ReLU | 0.0325 | 0.0369 | 0.0348 | 0.0457 | 0.0375 |
| Sine | Swish | **0.0314** | 0.0355 | **0.0338** | 0.0443 | 0.0363 |
| Gauss | ReLU | 0.0334 | 0.0382 | 0.0357 | 0.0472 | 0.0386 |
| Gauss | Swish | **0.0317** | 0.0360 | 0.0344 | 0.0452 | 0.0368 |
| Linear+Act | ReLU | 0.0335 | 0.0380 | 0.0350 | 0.0462 | 0.0382 |
| Linear+Act | Swish | **0.0317** | 0.0357 | **0.0337** | 0.0440 | 0.0363 |
| Identity | ReLU | 0.0337 | 0.0381 | 0.0354 | 0.0465 | 0.0384 |
| Identity | Swish | 0.0322 | 0.0365 | **0.0340** | 0.0446 | 0.0368 |
| None | ReLU | 0.0363 | 0.0424 | 0.0380 | 0.0514 | 0.0420 |
| None | Swish | 0.0330 | 0.0373 | 0.0348 | 0.0457 | 0.0377 |

where $\mathbf{MLP}(\cdot)$ is a 1-hidden-layer MLP with ReLU activation and layer normalization (Ba et al., 2016) applied before the activation. For aggregating the message, GNS used either mean or sum, so we tried both in our experiments. We found sum aggregation to perform better, and report results of sum aggregation in Table 1. After the messages are aggregated, GNS uses a learnable linear function to transform the node embeddings. Similar to ForceNet, we additionally apply a batch normalization on the node embeddings, which alleviates training instability and significantly improves performance. GNS uses a residual connection, where the computed node embeddings are added into the node embeddings from the previous layer. For the decoder, the GNS model uses a 1-hidden-layer MLP with ReLU activation. All the node embeddings and hidden units in the MLPs have the same dimensionality.

## C  HYPER-PARAMETERS

For training, we use the Adam optimizer (Kingma & Ba, 2015), with an initial learning rate of 0.0005. By default, we set the hidden dimensionality $D$ (*i.e.*, width) to 512, number of layers (*i.e.*, depth) to 5, batch size to 256. We train all the models for 500K iterations, which is equivalent to 1 epoch for the entire dataset[1] and takes 3 to 4 days on 16 Tesla V100 GPUs. All the parameters are initialized with Xavier uniform initialization (Glorot & Bengio, 2010). The learning rate is kept constant for the first 250K iterations, after which it is halved every 50K iterations. For batch sizes of 512, we double the initial learning rate (Goyal et al., 2017) and keep it constant for the first 150K steps, after which it is halved every 50K steps. We use the checkpoint with the best validation ID performance, and evaluate the saved model over all four validation sets. MAE over forces is used for the training loss. We will evaluate our models on the hidden test sets once the test server is ready.

For Gaussian and sine basis functions, we use $J = 50$, which gives an output dimensionality of $B = 350$. For Linear+Act, we set $B = 350$. For spherical basis, we use $L = 3$ and $S = 4$, which results in $B = 36(= 3^2 \cdot 4)$, and we set $J = 50$ for the internally used sine basis function. For encoding the input atomic node features, we first normalize each dimension to lie between 0 and 1, and adopt the same basis function as used for encoding the edge features. The exception is spherical basis that is specialized for 3D spaces, instead sine basis is used. We find that increasing $J$ and $L$ beyond the above values does not improve the performance, while significantly decreasing them worsens the performance.

## D  FULL ABLATION RESULTS

Here we provide detailed results of our ablation studies on ForceNet design as well as its training strategies.

---

[1]We do not observe much gain by training models longer than 1 epoch. This is probably because of the redundancy in data, *i.e.*, out of 130M data points, there are 650k unique atom configurations (ignoring the positional differences).

Table 6: Ablations on the message passing architecture of ForceNet.

| Ablation | Validation Force MAE (eV/Å) | | | | |
| --- | --- | --- | --- | --- | --- |
| | ID | OOD Ads. | OOD Cat. | OOD Both | Average |
| **ForceNet** | **0.0314** | **0.0348** | **0.0336** | **0.0433** | **0.0358** |
| (1) Only-dist | 0.0659 | 0.0670 | 0.0661 | 0.0802 | 0.0698 |
| (2) No-atomic-radii | 0.0319 | 0.0362 | 0.0342 | 0.0448 | 0.0368 |
| (3) No-node-emb | 0.0362 | 0.0413 | 0.0375 | 0.0498 | 0.0412 |
| (4) Only-$F_c$ | 0.0334 | 0.0371 | 0.0351 | 0.0455 | 0.0378 |
| (5) Edge-linear-BN | 0.0373 | 0.0427 | 0.0391 | 0.0520 | 0.0427 |
| (6) Node-linear-BN | **0.0315** | **0.0349** | **0.0334** | **0.0431** | **0.0357** |

Table 7: The effect of rotation augmentation and different weighting coefficients on fixed atoms while training ForceNet.

| Model | Rotation aug. | Weight on fixed atoms | Validation Force MAE (eV/Å) | | | | |
| --- | --- | --- | --- | --- | --- | --- | --- |
| | | | ID | OOD Ads. | OOS Cat. | OOD Both | Average |
| ForceNet | ✔ | 0.05 | **0.0314** | **0.0348** | **0.0336** | **0.0433** | **0.0358** |
| ForceNet | | 0.05 | **0.0318** | 0.0359 | 0.0343 | 0.0448 | 0.0367 |
| ForceNet | ✔ | 1 | 0.0367 | 0.0413 | 0.0388 | 0.0507 | 0.0418 |
| ForceNet | ✔ | 0 | 0.0324 | 0.0375 | **0.0340** | 0.0455 | 0.0374 |

**Full Results on ForceNet Designs**    First, we provide the full ablation results on ForceNet designs *for each* of the four validation sets. Table 5 shows the full ablation results on basis and activation functions, while Table 6 shows the full ablation results on the message passing architectures.

Overall, we observe the trends that are consistent with the averaged results in Figure 3 and Table 2. Specifically, from Table 5, we see that the combination of spherical basis functions and the Swish activation results in the best performance across the four validation sets. From Table 6, we see that the conditional filter convolution design gives superior performance compared to the more simplified architectures, except for (6) node-linear-BN, in which case the performance is comparable.

**Effect of Training Strategies**    Next, we provide results of ablation studies on our training strategies presented in Section 3.2 for a fixed model and optimization procedure. For the model, we use ForceNet with depth of 5, width of 512, with the spherical basis function and the Swish activation function. For optimization, we use the default configuration presented in Section 4.

The results are shown in Table 7. First, we see that rotation augmentation helps, especially for the three out-of-distribution validation sets. Second, we see that providing small reweighted supervision on fixed atoms is also helpful, significantly improving the validation performance (evaluated on *free atoms*) compared to the two baseline strategies: (1) uniform loss weighting (equally weighting the losses for fixed and free atoms) and (2) zero-loss weighting (ignoring losses on fixed atoms during training).

# E    DETAILS OF STRUCTURE RELAXATION

Structure relaxations are performed using a PyTorch implementation of the Atomic Simulation Environment's (*ASE*) (Hjo (2017)) L-BFGS optimizer. Relaxations were terminated when a max-absolute per-atom force of 0.01 eV/Å or 200 simulation steps, whichever comes first. All DFT calculations were performed in the *Vienna Ab Initio Simulation Package* (VASP) (Kresse & Hafner (1994); Kresse & Furthmüller (1996a;b)), all of which are popular packages within the computational chemistry and catalysis communities.

