# OpenReview forum: "ForceNet: A Graph Neural Network for Large-Scale Quantum Chemistry Simulation"
_ICLR.cc/2021/Conference — Reject_

### Official Review · AnonReviewer4 · 2020-10-28
**Relatively new application domain + nice results + ok technical contribution.**

**Rating:** 7
**Confidence:** 4

**Review:**

This paper presents ForceNet, a graph neural network, for estimating per-atom forces based on the 3D molecular structure. The authors argue that force-centric learning could be better in terms of prediction accuracy compared to energy-centric learning. As such, they focused on force-centric simulation and did several simple, but effective optimizations to make their model more accurate and easier to scale to large size. Experiments validate the effectiveness of ForceNet.

Overall, I think it is a nice application paper. It is easy to follow and gives clear reasoning behind their designs. Though many of the design options look simple and may not be strong enough from a machine learning perspective, these designs are tailored towards
large-scale quantum chemistry simulation and would be nice to showcase the potential of machine learning models, especially GNNs, to these application domains. From this perspective, I am happy with its contribution.

Questions:
1. Would you make your OC20 dataset open-source upon acceptance? This would be a big plus for this paper, as it will promote more ML experts to pursue this field.
2. Section 3.1.3 is a little bit vague. Why would Swish be a good choice for the activation function? Little background and insights were given.

---

> ### Author Response · Authors · 2020-11-21
> **Response to R4**
>
> Thank you for your positive feedback! Below are our answers to your questions.
>
> **Re: Will the OC20 dataset be released?**
>
> Yes, the dataset has been released and is publicly available.
>
> **Re: More background on Swish needs to be provided.**
>
> Thank you for your suggestion. We provide more background below, and we will add more descriptions in the final version.
>
> At a high level, the shapes of ReLU and Swish are quite similar (see e.g., https://ibb.co/zQjRQxm for the pictorial comparison): for a very small negative input, both the activation functions take the value of 0, while for a very large positive input, both the activation functions grow linearly. The two key differences are: (1) Swish is differentiable everywhere with no corner-like point at 0 like ReLU, and (2) Swish has non-zero activation value almost everywhere (unlike ReLU whose value is zero for all negative input values). Both of these characteristics make Swish suitable to model complex forces compared to ReLU, since (1) atomic forces behave smoothly, not in the piecewise-linear and corner-like manner (see Figure 5 of the Swish paper: https://arxiv.org/pdf/1710.05941v1.pdf), (2) modeling atomic forces accurately requires expressivity. ReLU would make all the negative input zero, not affecting the model output, while Swish does not have such an issue.
>
> Although we have not rigorously verified our above two claims, our experiments suggest the obvious gain by using Swish instead of ReLU. Moreover, in our preliminary experiments, we observed the performance of activation function was in the order of: swish > shifted softplus > softplus > leaky relu > relu >>> tanh = sigmoid (see their shapes at https://ibb.co/8YnX7DT). This ordering also suggests that our above two claims are fairly reasonable.

---

### Official Review · AnonReviewer1 · 2020-10-28
**Fast and accurate model with limited applicability**

**Rating:** 6
**Confidence:** 4

**Review:**

The paper propose a neural network force field that predicts atomic forces directly. This has the benefit of not requiring to differentiate an energy model and may be more flexible. The approach is well motivated and the paper is well structured and written. A strong point of the paper is the extensive discussion of model design choices.

A major weakness of the model is its lack of rotational covariance. In the presented application to relaxation of molecules on surfaces, this might be acceptable (since rotation only occurs in 2d here and relaxation path usually do not have a lot of structural variance), but for more flexible systems this will certainly lead to problems. I doubt that data augmentation can make up for this and the deviation of force predictions under rotation should be shown. In particular for MD simulations this might be a deal breaker, since the resulting model might not be energy conserving. For these reasons, the proposed approach might be of limited practical use beyond the demonstrated application. To prove otherwise, additional experiments with more flexible systems would be required.

Finally, the authors state: "However, practical models for real large-scale and complex problems remain out of reach." This is a bit of an overstatement as there exist ML force field approaches for practical problem ranging from nuclear quantum effect over reactions and spectroscopy to thermodynamics with thousands of atoms (e.g. refer to Table 1 in https://arxiv.org/pdf/2010.07067.pdf for an overview). For the particular problem of structure relaxation with ML, there is previous work by Shapeev et al (Phys. Rev. B, 2019) on accelerating crystal structure prediction.

Pros
------
- better and faster on the task than previous models
- extensive model selection and ablation studies

Cons
-------
- not energy conserving / rotational covariant
- thus probably limited applicability -> evaluate on a dataset with more variance and 3d rotation, e.g. MD trajectories


Update:
I read the response of the authors.

---

> ### Author Response · Authors · 2020-11-21
> **Response to R1**
>
> Thank you for your positive feedback! We have addressed R1’s major concern on rotation-covariance in the common response. In addition, R1 raises an important concern about the limited applicability of our ForceNet model to the conventional MD domains. Although we indeed designed the ForceNet model for the new application domain (i.e., OC20 dataset), we believe much of our architectural insights can be transferred to other similar simulation applications for which a large quantity of data is available. Examples include simulations of traffic systems, dynamics of sport players, and dynamics of weather, some of which may not have obvious physics knowledge to incorporate but are complex and have a large amount of data to train models. Moreover, we  believe designing a focused model for the new application domain is important for the two reasons listed below.
>
> (1) The new application is of high practical importance. The discovery of new materials within large design spaces is significantly bottlenecked by the time-consuming DFT calculations.
>
> (2) The new domain of material design presents a setting that is quite different from the conventional QM9 and MD datasets in many aspects, e.g., DFT data is generated with unprecedented quantity, the task centers on per-atom force prediction, and the z-axis is canonicalized with an infinitely-repeating structure along x,y-axis. As such, the optimal modeling strategy to the new domain is likely to be different from conventional molecule modeling techniques developed for the existing QM9 and MD datasets.
>
> In this paper, we explored and demonstrated a promising new strategy, where we train an expressive (conditional convolutional filter + basis function) and scalable (forward computation is linear w.r.t. number of edges unlike DimeNet, and no additional backprop is required) model on the large dataset.

---

### Official Review · AnonReviewer2 · 2020-10-29
**Report on ForceNet**

**Rating:** 5
**Confidence:** 5

**Review:**

##########################################################################
Summary:
This article presents the force fields predictor ForceNet. The authors show that ForceNet reduces the estimation error of atomic forces by 30% compared to existing ML models. Reconstructing molecular force fields is a very active and thriving field, hence this paper is highly relevant. In short, this article presents a very good model based a selection of very good methodologies and combining them in a clever way.

##########################################################################
Reasons for score:
Overall, I vote for not accepting, but if the authors address most of the comments I don't have a problem if it is accepted. As mentioned before, this problem is of high relevance and is likely to have a considerable impact in the physics and simulations community. Nevertheless, this is a non-energy conserving and non-rotationaly covariant force field, meaning that for most of the applications of a force field, this will provide unphysical results.

##########################################################################

Pros:
1. ForceNet reduces the estimation error of atomic forces by 30% compared to existing ML models.

2. It is based on strong and reliable previous results, which makes it a robust model.

##########################################################################
Comments:

-The authors state ”However, if successful, accurate and fast ML-based models may lead to significant practical impact by accelerating simulations from O(hours-days) to O(ms-s), which in turn accelerates applications such as catalyst discovery.” And “If successful, ML could be applied to problems such as catalyst discovery which is key to solving many societal…” These statements hint that this has not been done yet, which is a half truth. As exposed in the references cited in the paper, there are already very successful ML-methodologies in this field, but it has to be stressed that the successes come from applications in local domains. This mean that errors of sub-0.1 kcal/mol have been achieved for molecules already couple of years ago (Chmiela et al Nat Commun 9, 3887 (2018), Christensen et al J. Chem. Phys. 152, 044107 (2020), and Unke et al J. Chem. Theory Comput. 2019, 15, 6, 3678–3693). Hence, phrasing the manuscript like this, it could be misleading for the reader.
-The authors state “This is possibly because the force-centric models capture the dependency of atomic interactions on atomic forces more explicitly than the energy-centric models.” This topic has been extensively analysed by Prof. Müller’s group (Sci. Adv. 3 (5), e1603015, 2017; Nat Commun 9, 3887 (2018)) and recently by Prof. von Lilienfeld (Mach. Learn.: Sci. Technol. 1 045018, 2020). In short, learning forces is equivalent to learn linearisation of the energy surface which is much more informative.
-MLP is not defined prior usage.
-The directed message e_{st} is a vector or scalar? In some parts appeared just as e_{st} and on other as \textbf{e}_{st}.
-The atomic radii is a given physical value or is it a learning parameter?
-The authors state: “As we demonstrate in Section 4, the replacement of ReLU with Swish consistently and significantly improves the predictive accuracy while maintaining scalability across all choices of basis functions.” This is not relevant, since it is a direct statement. A more interesting comparison would be to shifted tanh function.

-From my point of view, the main downside of this article is the fact that the ForceNet is neither exactly covariant not energy conserving.

---

> ### Author Response · Authors · 2020-11-21
> **Response to R2**
>
> Thank you for your detailed and insightful feedback. We have addressed your main concern regarding rotation-covariance and energy-conservation in the common reply. Here we address your other concerns.
>
> **Re: There are already successful applications of ML in the field such as (Chmiela et al Nat Commun 9, 3887 (2018), Christensen et al J. Chem. Phys. 152, 044107 (2020), and Unke et al J. Chem. Theory Comput. 2019, 15, 6, 3678–3693).**
>
> This is a very good point. We agree that successful ML applications have been done in the past to various extents. However, as R2 has mentioned, much of these models have been limited to very specific applications and narrow domains (small organic molecules, specific chemistries, etc.). Additionally, existing ML models in this domain can scale poorly with the number of elements in the system (Behler-Parinello for instance), making their application to OC20 quite impractical with 55 elements. Despite the success of these models to very specific, densely sampled applications, their success to large, sparsely sampled datasets (OC20) has not been demonstrated. A successful ML model for such a dataset would enable large-scale catalyst discovery across a broad range of elements and surfaces. We will clarify this phrasing in the manuscript as you have suggested.
>
> **Re: The effectiveness of the force-centric model has been extensively explained in the literature, e.g., (Sci. Adv. 3 (5), e1603015, 2017; Nat Commun 9, 3887 (2018)).**
>
> Thank you for your pointers. We will cite these works when making our argument. To clarify the potential misunderstanding, the energy-centric baseline models also utilize force information in their learning by directly minimizing the discrepancy between the energy derivatives and the ground-truth forces. Therefore, the force supervision that the energy-centric model and force-centric model get is exactly the same (it’s not that the energy-centric model is only given the energy as the supervision). Given this, we have empirically demonstrated that the force-centric approach could be more promising than the energy-centric approach in predicting atomic forces, which is a highly non-trivial finding. We are going to further clarify this in the final version of the paper.
>
> **Re: The directed message e_{st} is a vector or scalar?**
>
> Thank you for noticing the typo. e_{st} is always a vector, containing distance information and angular information. We will fix the typo in the final version.
>
> **Re: The atomic radii is a given physical value or is it a learning parameter?**
>
> It is a given physical value retrieved from https://mendeleev.readthedocs.io/en/stable/index.html and is fixed throughout learning.
>
> **Re: The authors state: “As we demonstrate in Section 4, the replacement of ReLU with Swish consistently and significantly improves the predictive accuracy while maintaining scalability across all choices of basis functions.” This is not relevant, since it is a direct statement. A more interesting comparison would be to shifted tanh function.**
>
> We are not quite sure if we understand the sentence “This is not relevant, since it is a direct statement”. We do believe the comparison is relevant since ReLU and Swish are quite similar in shape (see their pictorial comparison at https://ibb.co/zQjRQxm), and it is striking that the seemingly small differences (differentiably smooth, and non-zero negative part) consistently makes a significant impact on the final performance. Additionally, in our preliminary experiments, we have tried both tanh and sigmoid but they performed much worse than both ReLU and Swish possibly due to the vanishing gradient problem.

---

### Official Review · AnonReviewer3 · 2020-10-31
**GNN + Continuous Convolution for Quantum Chemistry Simulation**

**Rating:** 7
**Confidence:** 5

**Review:**

**Summary**
This paper proposes ForceNet which is capable of capturing highly complex and non-linear interaction of atoms in 3D molecular space in order to accurately predict atomic forces. On the outset, the ForceNet framework seems extension of recent work on physics-based simulator for fluid dynamics using deep network called GNS (Sanchez-Gonzalez et al, 2020) but with carefully designed message passing network. Unlike GNS, the ForceNet computes messages using specifically designed continuous filter convolution. Inspired by SchNet, the filter weights in continuous filter are dynamically computed by first lifting edge feature input to new basis space followed by multi layer MLP. However, unlike SchNet, the filter computation is conditioned on the source and target node features as well as the 3D angular information.

Apart from this, ForceNet deploys spherical harmonic basis functions and smooth activation function - Swish, to realised better accuracy. Experimental evaluation on large-scale catalyst dataset OC20 shows that ForceNet reduces force estimation error by 30% in comparison to recently popular ML models for quantum chemistry simulation. When employed for quantum chemistry simulation, wherein DFT forces are replaced with ForceNet computed forces, it converges to similar energy structure as DFT in $10^3 \times$ lesser time.

**Quality**
The paper is very well written and easy to follow. The description of their model, experimental setup, comparison to existing baselines and their architectural choices are precisely described (as well as evaluated). Although their largest but best model is $3\times$ slower than existing work,  it produces highly accurate force prediction.

**Originality**
As mentioned above, this work combines best of two world proposed for quantum chemistry simulation in the past - GNN based framework and continuous convolution. From the outset, the architecture seem to me simple but effective extension of SchNet work with modifications in interaction layer such as,
1. Use of directional vector and atomic radii for feature input to basis function instead of mere scalar distances.
2. Convolution filter generation conditioned on source and target node features
3. Inclusion of self-message $m_t$ and scalar multiplier $\alpha$ and
4. Different basis and activation function

ForceNet achieves this by trading off rotation-invariant property while leveraging random rotation data augmentation to achieve rotation covariant.

**Significance**
The carefully designed message passing computation will motivate future work to design better layers for targeted application. Moreover, the speed + accuracy achieved on DFT task is very encouraging for future deployment of ML model for quantum chemistry simulation.

**Clarity**
1. Like previous works, can you compare errors w.r.t. energy output ?
2. SchNet uses shifted softplus as smooth activation. Ofcourse the motivation there was twice differentiability. Have you experimented ForceNet with shifted softplus as well as other non-smooth activation such as Leaky ReLU etc. ?
3. For Table 2, can you share results obtained by only omitting $m_t$.
4. Have you tried spherical bessel basis as in DimeNet ? Comparison to this was missing under basis function choices.

---

> ### Author Response · Authors · 2020-11-21
> **Response to R3**
>
> Thank you for your positive feedback! Below are the answers to your questions.
>
> **Re: Do we have the energy output?**
>
> We only predict forces in our work. We leave the investigation of energy prediction for future work, for which we could train a separate model.
>
> **Re: What is the performance of other activation functions, such as shifted softplus and leaky relu?**
>
> That’s a great question. In our preliminary experiments, we did try different activations. We found the performance was in the order of swish > shifted softplus > softplus > leaky relu > relu >>> tanh = sigmoid (see their shapes pictorially at https://ibb.co/8YnX7DT). We generally found that non-piecewise-linear and non-zero activation in the negative part are helpful. We also found saturating activations, such as tanh and sigmoid, are difficult to train, possibly because of the vanishing gradient problem. Based on our preliminary experiments, we picked the swish activation and conducted the large-scale empirical study in the paper. We will add a short description of the above to the final version of our paper.
>
> **Re: Can you show the result without self-message m_t?**
>
> Thank you for your suggestion. We have conducted the suggested experiment, and the result is shown below.
>
> **Force MAE (the lower, the better)**
>
> Val: without m_t / with m_t (ForceNet)
>
> Avg: 0.0362 / 0.0358
>
> ID: 0.0315 / 0.0314
>
> Ads: 0.0353 / 0.0348
>
> Cat: 0.0338 /  0.0336
>
> Both: 0.0438 / 0.0433
>
> We see that removing m_t does not affect the performance much, implying that the self-message m_t is not essential to ForceNet’s performance. This is expected because (1) the number of learnable parameters associated with m_t is extremely small (only D per layer), and (2) m_t is not directly related to capturing neighboring 3D structures via inter-node message passing. This ablation result accords well with our claim that ForceNet’s performance mainly comes from the expressive inter-node message passing that allows the model to accurately capture 3D structure. We will add this result and discussion in our final version.
>
> **Re: Comparison with the spherical Bessel in DimeNet.**
>
> Our basis function needs to encode the full 3D structure (r, theta, phi), unlike the spherical bessel basis function in DimeNet that only encodes partial angular information (r, theta). Thus, DimeNet’s basis function is not directly applicable to our setting. Our spherical harmonics basis function can be considered as a natural generalization of DimeNet’s 2D basis function to the full 3D space.

---

### Author Response · Authors · 2020-11-21
**Common Response to Reviewers**

We thank the reviewers for their extensive and insightful feedback. Overall, we are glad that all the reviewers find our ForceNet model to be effective with extensive and insightful experiments, and the application of material discovery to have high potential for GNNs to demonstrate significant real-world impact.

In the common response here, we address the important concern raised by the reviewers (especially, R1 and R2) that ForceNet does not incorporate the physical rules in its architecture design (not energy conserving and rotation covariant). Prior models such as SchNet and DimeNet do enforce those physical rules in their model architectures. However, in practice, we find these prior models to perform much worse than the ForceNet model in our domain of interest, large-scale quantum chemistry simulations. This raises the interesting question of whether explicit enforcement of these rules is necessary. We see our work as offering an exciting demonstration that explicit rules may not be necessary if large expressive models are trained with massive data---a regime that has not been explored much in the GNN literature.

Furthermore, interestingly, we find ForceNet does capture some basic physical rules, even if they are not explicitly enforced. To demonstrate this, we measured the force MAE as a structure is rotated to determine whether the ForceNet-large’s prediction was covariant to rotation. We randomly sampled 200 validation systems, and for each system, we rotate the system for {0, 18, 36, …, 360} degrees along the z-axis. We then pass the rotated system to ForceNet-large to predict atomic forces, and finally we rotate back the predicted forces to calculate the force MAE on the system. If the model is rotationally covariant we would expect the force MAE to remain constant across rotations. As shown in the figure below, we observe the force MAE is actually stable with respect to rotations, which implies the model does learn some degree of rotation-covariance.

**Figure** (generated via imgbb and fully anonymized):
https://ibb.co/Dw1Dvx3

Moreover, any force-centric model, including ForceNet, can be made more robust to rotation by averaging predictions over differently rotated systems (where the prediction needs to be rotated back before the average is taken). By doing so, an even more rotation-covariant ForceNet model can be obtained. We leave further investigation of this for future work.

Overall, we demonstrate that expressive models can outperform those with explicit enforcement of physical rules. While strict adherence to physical rules may be a desirable property, it may overly constrain the network during optimization, resulting in reduced overall accuracy. Additionally, enforcing physical rules often results in much more complex and computationally expensive models that are hard to scale to the larger atomic systems in the OC20 dataset (e.g., DimeNet as well as the very recent physics-based neural models such as differentiable DFT [1] and FermiNet [2]). In our work, we directly train the expressive model on massive data, through which the model learns some basic physical rules as well as potentially much more complex ones. It is exciting future research to explore how physics knowledge can be effectively incorporated into ForceNet to further improve the performance. We thank the reviewers for making this important point, which we will further clarify in the final version of the paper.

[1] https://arxiv.org/pdf/2009.08551.pdf
Performance was demonstrated only for 1D system, and scaling beyond it is challenging.

[2] https://journals.aps.org/prresearch/abstract/10.1103/PhysRevResearch.2.033429
Performance was demonstrated for atoms and small molecules. The model scales with the number of electrons in the system; thus, scaling to larger molecules is challenging.

---

### Decision · Program_Chairs · 2021-01-07
**Final Decision**

**Decision:**

Reject

**Comment:**

The model presented here may be of use to others in running quantum chemistry simulations, and it may well lead to new advances, but the authors did not sufficiently address the key concerns around the model not being energy conserving and rotation covariant. The approach proposed could be learning such physical rules, and the authors in their general response provide some very preliminary evidence for this, but a much more thorough discussion with a full range of experiments is needed. ICLR is a broad conference where non-experts who may have never heard of DFT simulations must parse this work and decide on if/how to follow up. This is a critical missing piece for anyone that wants to do so.

The above is exacerbated by the fact that the work is not well-situated against prior work as pointed out by two reviewers. Together these two issues conspire to make understanding this model, the contribution of the work, and what followup is possible, untenable for an ICLR reader. For example, one would not be able to surmise from the manuscript and its brief discussion of rotation covariance that this is likely to result in ForceNet having limited applicability to other DM problems; which one reviewer pointed out and the authors generally agreed with. While the authors respond that perhaps the architecture itself may be useful for other applications, why this would be and what the specific advantage of the current model relative to the state of the art in those fields is unclear.